# Mycolactone A vs. B: Multiscale Simulations Reveal the Roles of Localization and Association in Isomer-Specific Toxicity

**DOI:** 10.3390/toxins15080486

**Published:** 2023-08-02

**Authors:** John D. M. Nguyen, Gabriel C. A. da Hora, Jessica M. J. Swanson

**Affiliations:** Department of Chemistry, University of Utah, Salt Lake City, UT 84112, USA; johnny.nguyen@utah.edu (J.D.M.N.); gabriel.hora@utah.edu (G.C.A.d.H.)

**Keywords:** mycolactone, Buruli ulcer, ER membrane, Sec61 translocon, molecular dynamics, membrane permeation, isomeric specificity

## Abstract

Mycolactone is an exotoxin produced by *Mycobacterium ulcerans* that causes the neglected tropical skin disease Buruli ulcer. This toxin inhibits the Sec61 translocon in the endoplasmic reticulum (ER), preventing the host cell from producing several secretory and transmembrane proteins, resulting in cytotoxic and immunomodulatory effects. Interestingly, only one of the two dominant isoforms of mycolactone is cytotoxic. Here, we investigate the origin of this specificity by performing extensive molecular dynamics (MD) simulations with enhanced free energy sampling to query the association trends of the two isoforms with both the Sec61 translocon, using two distinct cryo-electron microscopy (cryo-EM) models as references, and the ER membrane, which serves as a toxin reservoir prior to association. Our results suggest that mycolactone B (the cytotoxic isoform) has a stronger association with the ER membrane than mycolactone A due to more favorable interactions with membrane lipids and water molecules. This could increase the reservoir of toxin proximal to the Sec61 translocon. In one model of Sec61 inhibited by mycolactone, we find that isomer B interacts more closely with residues thought to play a key role in signal peptide recognition and, thus, are essential for subsequent protein translocation. In the other model, we find that isomer B interacts more closely with the lumenal and lateral gates of the translocon, the dynamics of which are essential for protein translocation. These interactions induce a more closed conformation, which has been suggested to block signal peptide insertion and subsequent protein translocation. Collectively, these findings suggest that isomer B’s unique cytotoxicity is a consequence of both increased localization to the ER membrane and channel-locking association with the Sec61 translocon, facets that could be targeted in the development of Buruli Ulcer diagnostics and Sec61-targeted therapeutics.

## 1. Introduction

Buruli ulcer is a neglected tropical disease caused by *Mycobacterium ulcerans* (*M. ulcerans*) that leads to large necrotic skin ulcerations that surprisingly demonstrate no pain and inflammation [1,2,3]. These pathogenic effects are caused by a macrolide exotoxin produced by the bacteria known as mycolactone. The toxin accumulates in the extracellular matrix of *M. ulcerans* and is secreted in vesicles [4]. Once delivered to host cells, mycolactone induces cytotoxic and immunosuppressive effects by disrupting several cellular processes, such as cytoskeletal organization, protein production and signaling cascades [2,4,5]. The mycolactone-containing vesicles have been suggested to be important for toxin delivery as purified mycolactone was found to be significantly less cytotoxic than purified vesicles from *M. ulcerans* [4]. Structurally, mycolactone consists of a 12-membered lactone ring with two highly unsaturated chains (Figure 1). The southern chain has been shown to play a key role in the toxin’s cytotoxicity, first through mycolactone variants that differ in bioactivity that possess the same northern chain and lactone ring structure but vary in the southern chain [2,6,7]. Additionally, it has been reported that synthetic mycolactone subunits lacking the northern chain partially retained immunosuppressive activity, while subunits lacking the southern chain or both chains were inactive [8]. Although these structure-activity relationships have been reported, the underlying mechanisms remain unknown.

Mycolactone exists primarily as two cis/trans stereoisomers, designated as mycolactone A and mycolactone B, respectively; they are in dynamic equilibrium with a 60:40 ratio [9]. These two isoforms differ in a rotation around the toxin’s 4–5 double bond in the southern chain (Figure 1). The bioactivity of the two isomers was studied by synthesizing analogs that closely resemble the natural compounds but with slight structural variations, while preserving the cis/trans aspect of the southern chain. Results showed that the B analog was significantly more cytotoxic, suggesting that mycolactone B is the primary virulence factor of Buruli ulcer [10]. Interesting possible explanations of this isomeric specificity include differences in adsorption, cellular localization and/or host target interactions.

Mycolactone was initially believed to enter the host cytosol following passive diffusion through the cell membrane [11]. However, computational studies initially predicted [12], and experimental work showed that, the toxin strongly interacts with cellular membranes [13]. Nitenberg et al. [13] studied mycolactone’s interaction with membranes using Langmuir monolayers and reported that the toxin interacts with membranes at very low concentrations and disturbs lipid organization. Our previous work quantified a strong free energy of association between mycolactone and phospholipid membranes and revealed the toxin’s preference for the interfacial region of bilayers below the lipid headgroups [12,14]. Most recently, we demonstrated that mycolactone B has a stronger association free energy with the ER membrane than with a model plasma membrane [15]. Moreover, the toxin reorganizes the plasma membrane to reflect the more disordered ER-like composition locally. Given recent work demonstrating the role of lipid order trafficking proteins [16], it is possible that the toxin’s preferential lipid interactions influence its cellular localization. Given these findings, it is likely that what was previously believed to be the toxin’s localization in the cytosol was more likely the association of mycolactone with the ER membrane.

Mycolactone acts by invading host cells and interacting with multiple intracellular targets, resulting in various cellular effects. One such target is the Angiotensin II receptor (AT2R), a key player in signal transduction among neurons responsible for pain perception. By binding to this receptor, the toxin has been suggested to initiate a signaling cascade that ultimately leads to hyperpolarization of neurons, impairing their ability to transmit pain signals and resulting in analgesia [17,18]. Another potential mycolactone target is the Wiskott–Aldrich syndrome protein (WASP) and its neuronal version N-WASP. WASP/N-WASP are autoinhibited regulatory proteins that activate actin branching and polymerization, thereby influencing the cytoskeleton’s structural integrity and functions such as movement and adhesion [19,20]. Mycolactone was shown to activate WASP in cell-free assays and to colocalize with WASP in HeLa cells, suggesting it could play a role in the observed defective adhesion and migration [21,22]. However, a recent study reported that the structurally unrelated Sec61 inhibitor Ipomoeassin F induced the same morphological effects as mycolactone in epithelial cells, including impaired migration, suggesting that the WASP interaction may not be the cause of altered adhesion/migration [23]. This is consistent with other studies showing that mycolactone’s main cytotoxic and immunosuppressive effects result from its inhibition of the Sec61 translocon [6], a membrane-embedded protein complex (Figure 2) that translocates precursor polypeptides into the ER for processing [1]. The toxin binds to the pore-forming Sec61α subunit of the protein and strongly inhibits its ability to translocate polypeptides. This reduces the cell’s ability to produce several secretory and transmembrane proteins, which leads to multiple downstream consequences, such as immunomodulation and cytotoxic effects [1,24]. Single amino acid mutations in Sec61 have been identified that confer cells broad resistance to mycolactone’s cytotoxic and immunomodulatory effects while not affecting the protein’s functionality [6].

Two cryo-electron microscopy (cryo-EM) structures of Sec61 inhibited by mycolactone have been reported. They differ in the position and orientation of the bound toxin and structure of Sec61. In the first, reported by Gérard et al. [25] (PDB ID: 6Z3T), the toxin is located between transmembrane segments (TMs) that make up a key gating element referred to as the lateral gate (Figure 2A). The lateral gate consists of TMs 2b, 3, 7 and 8, which separate to open the channel laterally and allow peptides access to the ER membrane environment [27]. Gérard et al. [25] showed that mycolactone wedges open the cytosolic side of the lateral gate, trapping the translocon in a partially active state. This conformation of Sec61 inhibited by mycolactone is structurally similar to recently reported cryo-EM structures of the translocon inhibited by a CADA derivative, CK147 [28], and cotransin analogue, KZR-8445 [29], despite the inhibitors being bound in different sites in each case. The toxin binds in a conformation where the southern chain protrudes into the lipid tail region of the ER membrane, while the northern chain is embedded in the protein. This proposed binding site is lined with a hydrophobic patch of residues (V85, L89, I179) that contact mycolactone and are thought to serve as an initial interaction site for hydrophobic signals to facilitate signal peptide binding [30]. Gérard et al. [25] found that mutations of the hydrophobic patch residues do not confer mycolactone resistance. However, the authors explained that it is possible that mutations of these residues could render the translocon non-functional. Thus, it can be suggested that mycolactone’s cytotoxic effects are a result of the toxin blocking hydrophobic patch residues to prevent signal peptide binding and subsequent peptide translocation.

The second cryo-EM structure was recently reported in a study by Itskanov et al. [26] (PDB ID: 8DO0), wherein the authors obtained near-atomic-resolution cryo-EM structures of the human Sec61 translocon bound by seven small molecule inhibitors. Although each inhibitor in the study possessed distinct chemical structures, they all interact with hydrophobic side chains of three key gating elements: the plug, the lateral gate and the pore ring (Figure 2). The plug is a short helix that must be displaced to allow translocating peptides access to the ER lumen. The pore ring is a constriction of six hydrophobic residues (I81, V85, I179, I183, I292, and I449) that translocating peptides pass through; it is gated by movement of the plug [27]. Each of the inhibitors studied by Itskanov et al. [26] interact with TMs 2b and 3, which form one side of the lateral gate, as well as TM 7, forming the other side. Given that concerted movement of the plug and lateral gate is required for the insertion of an incoming signal sequence and subsequent translocation, this coordination of interactions likely blocks signal insertion. In addition to hydrophobic interactions with the three gating elements, the authors also found that all of the inhibitors interact with a polar cluster of residues from the lateral gate, which were confirmed to be key for binding affinity by mutational analysis of yeast equivalents [26]. Taken together, Itskanov et al.’s [26] findings suggest that all seven tested Sec61 inhibitors act by locking the lateral and lumenal gates of the channel to prevent dynamics required for translocation.

Comparing the two structures, the toxin is bound lower in the lateral gate TMs in the Itskanov model [26] relative to the Gérard [25], and in the reverse orientation with the southern chain embedded in the protein while the northern chain is exposed to the membrane environment (Figure 2B). The difference in binding sites may potentially be attributed to differences in model preparation. Gérard’s structure [25] was obtained using canine microsomes containing ribosome-translocon complexes in native lipids; approximately 300 nM of mycolactone was added while the Sec61 translocon remained in the membrane environment prior to extraction of the toxin-bound complex. In contrast, Itskanov et al. [26] used a chimeric translocon consisting of human transmembrane domains and yeast cytosolic domains. Instead of being ribosome bound, the chimeric complex was combined with allosteric activators Sec62 and Sec63 in order to stabilize the translocon in an open state. Moreover, 100 µM mycolactone was introduced to the chimeric translocon after the protein was purified and integrated into a non-lipid peptidisc. Among these differences in the preparation of the toxin-Sec61 complexes, the use of a membrane-embedded complex versus a non-lipid peptidisc is particularly noteworthy. Considering our previous studies [14,15], which demonstrate the toxin’s strong affinity for lipid membranes, it is likely that mycolactone reaches the translocon through the lipid membrane environment in Gérard’s model [25]. In contrast, the interaction between the toxin and the peptidisc in Itskanov’s preparation [26] is unknown. If mycolactone exhibits a weak affinity for the peptidisc helices surrounding the transmembrane region of the protein complex, the toxin may access the translocon pore from the aqueous medium.

In this work, we query the molecular mechanisms underlying mycolactone’s bioactivity by comparing the interactions of the two isomers with a model ER membrane and with the Sec61 translocon. We use MD combined with Transition-Tempered Metadynamics (TTMetaD) enhanced sampling in order to calculate the permeation free energy profiles and characterize the association of mycolactone A and B with a model ER membrane, aiming to investigate the role of localization in the toxin’s pathogenicity. The ER membrane was chosen since it is the likely reservoir for the toxin prior to its association with the Sec61 translocon. We also use MD to model membrane-embedded mycolactone A/B-Sec61 complexes using both Gérard et al.’s [25] and Itskanov et al.’s [26] cryo-EM models as references to identify the existence of interactions and/or dynamics that may contribute to isomeric specificity and predict the inhibition mechanism in both models. We first present permeation free energy profiles, which show a stronger affinity of mycolactone B for the ER membrane than mycolactone A. This has interesting ramifications for both cellular localization and competition between the isomers for the translocon. We find that isomer B has a more open structure compared to the A isomer’s more compact structure; this open structure allows it to better interact with the membrane lipids and water molecules while in the membrane. Comparing simulations of the toxin-Sec61 complexes, in Gérard’s model [25], we find that mycolactone B’s more open structure also allows it to better interact with the hydrophobic patch residues, suggesting an increased ability to block this region from signal peptides. In contrast, the more compact structure of the A isomer appears to limit its ability to fully enter the binding site to interact with the hydrophobic patch. In Itskanov’s model [26], we find that mycolactone A is unable to interact with the plug domain and TMs 7 to the extent that mycolactone B does; this suggests that isomer B has an increased ability to lock the plug domain and TMs 2, 3 and 7 in place to inhibit translocation. We also find that mycolactone B has more interactions with the pore ring and is more inserted into the constriction, which may result in a stronger binding affinity and increased ability to occlude the pore from peptides. Additionally, polar interactions are better satisfied with mycolactone B bound via interactions with the polar cluster residues of the lateral gate and water molecules. Lastly, the isomer B-bound complex adopts a more closed conformation on the cytosolic side of the translocon, which has been suggested to be linked to increased inhibition efficacy.

## 2. Results

### 2.1. Endoplasmic Reticulum Membrane Association

Two-dimensional free energy profiles (potentials mean of force; 2D-PMFs) for the permeation of mycolactone A and mycolactone B through the ER membrane are shown in Figure 3. The two collective variables (CVs) that were tracked are the permeation depth, tracked via the z-distance between the center of mass (COM) of the lactone ring and that of the membrane, and the orientation, tracked via the angle between the vector connecting the northern and southern chains’ hydroxyl groups and the membrane normal [15]. The black lines track the minimum free energy paths, which are the most likely paths taken by the isomers through the membrane. Inserted snapshots (Figure 3D,E) show the dominant configurations in minima along the minimum free energy paths. In these minima, the polar hydroxyl groups at the ends of each chain hydrogen bond with oxygen atoms from lipid headgroups, glycerol and/or water, while the macrolide ring and aliphatic portion of the southern chain interact with the hydrophobic lipid tails. This is consistent with our previous computational work, where we studied mycolactone B’s interaction with other model membranes [12,14,15].

The minimum free energy paths of both isomers through the ER membrane are similar: the toxin first adopts conformations that optimize polar and nonpolar interactions at the interfacial region of the membrane below the lipid headgroups, and then migrates from one leaflet to the other by swapping its polar interactions with lipid headgroups of one layer to lipid headgroups of the opposite leaflet. However, while associating with the membrane, it appears that isomer A’s compact structure has fewer favorable interactions with the membrane and more intramolecular interactions (Figure 3D), while isomer B’s more open structure demands more interactions with the bilayer and/or water (Figure 3E). There is also a difference in the binding affinity, with mycolactone B having a more favorable interaction with the ER membrane than mycolactone A. This is more evident in the corresponding one-dimensional free energy profiles (Figure 4). Additionally, there is a difference in the membrane-associated conformational ensemble; mycolactone A has a broader angle CV distribution than mycolactone B (Figure 3). The free energy landscapes in Figure 3A,B were validated by analyzing the distribution of the CVs in unbiased simulations and found to be consistent with mycolactone A exhibiting a broader angle distribution (Appendix A). The extended regions in the angular distribution correspond to configurations where only one of the hydroxyl-containing chains is interacting with the lipid head groups or glycerol, while the other is buried in the lipid tails, satisfying polar groups with intramolecular interactions (Figure 3D). This observation suggests that the interactions between mycolactone A’s hydroxyl tail ends and the lipid headgroups are more transient, whereas mycolactone B’s are more consistent.

To further query how the observed difference in binding affinities arises from the structural differences between the isomers, we turn to interaction energies. With a cis double bond in the southern chain, mycolactone A is more compact, which may shift interactions with its environment to interactions with itself. To test this, the GROMACS tool gmx_energy was used to calculate the average interaction potential energies between the toxin and the ER membrane and water molecules, as well as the intramolecular toxin interactions (Table 1). Energy values were calculated for configurations where mycolactone is in the membrane hydrophobic core (−2.0 < *Z* < 2.0 nm). From this analysis, it appears that while associating with the membrane, mycolactone B has a stronger interaction with the lipids and water molecules, while mycolactone A has stronger intramolecular interactions.

We further evaluated the interactions between the toxin and water molecules by calculating the probability distribution of the number of water molecules in contact with the toxin as a function of the z-component distance between the lactone ring and the center of the membrane (Figure 5). A contact is defined as any distance between an oxygen atom of water and any oxygen atom of mycolactone less than or equal to 3.04 angstroms. Our previous work [14,15] has shown that mycolactone B coordinates with water molecules during association with pure DPPC and model ER and plasma membranes to satisfy polar interactions as the toxin moves into hydrophobic regions of the bilayer. Here, both mycolactone A and B can interact with water molecules while permeating through the ER membrane, even while near the center of the membrane. However, mycolactone B is more likely to coordinate with more water molecules than mycolactone A in the ER membrane. This, again, is due to mycolactone A’s more compact structure, increasing its intramolecular hydrogen bonding instead of interacting with water molecules. These differences in water coordination likely contribute to mycolactone B’s stronger affinity for the ER membrane compared to isomer A.

Additionally, to investigate each isomer’s effect on the ER membrane’s physical properties, lipid tail order parameters were calculated for lipids near the toxin (within 5 angstroms) in unbiased simulations. The calculated order parameters were compared to tail order parameters of an ER membrane-only simulation (Appendix A). Both mycolactone A and B were found to disrupt lipid tail order to a comparable degree.

### 2.2. Sec61-Toxin Complex 1

Given that the toxin’s cytotoxic effects have been strongly linked to its interaction with the Sec61 translocon, we next evaluated differences in the association of the two isomers. Both Gérard’s [25] and Itskanov’s [26] cryo-EM structures were evaluated since it is not yet known which is biologically more relevant.

In the first set of A/B complexes, the initial structure was kindly provided by Gérard et al. [25] from their equilibrated MD simulation of the mycolactone B-Sec61 complex embedded in a pure POPC membrane. An initial structure of the mycolactone A-Sec61 complex was obtained by replacing the toxin’s 4–5 double bond (Figure 1) parameters with single bond parameters and allowing the toxin to convert to the A isoform (further described in Methods). Both isomers were oriented with their northern chain inserted into the core of the translocon while the southern chain protrudes out into the hydrophobic lipid tail region of the bilayer. This orientation was proposed to be the most likely binding mode by Gérard et al.’s [25] MD simulation data. Each complex was simulated for 3 microseconds and the last microsecond was used for analysis. Both isomers remained bound to the translocon throughout the simulations.

Contact analyses between isomers A/B and the Sec61 hydrophobic patch residues were conducted to evaluate each isomer’s interaction with this key region of the protein (Figure 6). A contact was defined as any distance between heavy atoms less than or equal to 4.5 angstroms. Mycolactone B was found to have significantly more contacts. It appears that the cis double bond in the southern chain of mycolactone A prevents it from fully entering the binding site. In contrast, mycolactone’s B trans double bond straightens the southern chain more and allows better access to the binding site to interact with the patch residues (Figure 7). Mycolactone B’s increased interaction may prevent incoming signal peptides from interacting with patch residues to initiate translocation. In contrast, with mycolactone A displaced further from the binding site, signal peptides might still be able to interact with the patch residues even while the A isomer is bound.

Following equilibration, there was a clear difference in the conformational ensembles of Sec61 with the two isomers bound. In the mycolactone B complex, the lateral gate exhibits a wedged open conformation on the cytosolic side, consistent with the observations made by Gérard et al. [25]. On the other hand, the mycolactone A complex shows a relatively more closed conformation of the lateral gate (Figure 8). This difference is likely attributable to the A isomer’s inability to fully enter the binding site and effectively separate TMs 2 and 7.

### 2.3. Sec61-Toxin Complex 2

In the second set of A/B complexes, the isomers were bound to a region of Sec61 that was identified to be mycolactone’s binding site by Itskanov et al. [26] via cryo-EM. Since only mycolactone B was modeled into the cryo-EM density, mycolactone A was docked as described in Methods. Once bound, both isomers were oriented with their southern chain inserted into the core of the translocon while the northern chain protrudes out into the hydrophobic lipid tail region of the bilayer. Each complex was simulated until the root-mean-squared deviations (RMSDs) of the protein backbone stabilized (1.2 and 1.5 microseconds for mycolactone A and B, respectively) (Appendix A); the last 300 nanoseconds were used for analysis. Both isomers remained bound to the translocon throughout the simulations.

We first analyzed interactions with the three key gating elements: the plug, the lateral gate and the pore ring. Contact analyses were performed between mycolactone isomers A/B and residues of each gating element. Once again, a contact was defined as any distance between a heavy atom of the toxin and a heavy atom of the protein less than or equal to 4.5 angstroms. Mycolactone B was found to have several contacts with the plug and both sides of the lateral gate (TMs 2b and 3 on one side and TM 7 on the other), while mycolactone A predominantly interacts with just TMs 2b and 3 (Figure 9A–C and Figure 10). The A isomer’s lack of interactions with the plug and TM 7 likely results in a weaker ability to cement the plug and two sides of the lateral gate together at their interface to restrict dynamics and inhibit peptide translocation. Mycolactone B was also found to have more contacts with the pore ring as it intercalates into the crescent-shaped constriction, whereas mycolactone A is bound further away (Figure 9D and Figure 10). This difference in interaction with the pore ring may result in a stronger binding affinity of isomer B and a possible increased ability to block the pore from peptides.

To evaluate the key polar interactions shared by all inhibitors and demonstrated as important for binding via mutational analysis by Itskanov et al. [26], the number of contacts between oxygen atoms of the toxin and oxygen and nitrogen atoms of the polar cluster residues of the lateral gate (Q127 and N300) was calculated. Mycolactone B was found to have more polar interactions with the polar cluster residues than mycolactone A, which likely contributes to a stronger binding affinity of the B isomer (Figure 9E and Figure 10). Interestingly, effectively all the contacts calculated for both isomers belonged to interactions with N300 (100% and 99.7% for mycolactone A and B, respectively), suggesting that N300 is more crucial for the binding affinity than Q127. This is consistent with Itskanov et al.’s [26] mutational analysis, which revealed that an N300L mutation conferred stronger resistance than a Q127L mutation to Sec61 inhibitors (cotransin, decatransin and ipomoeassin F). In addition to polar interactions with the polar cluster residues, Itskanov et al.’s [26] mycolactone-bound structure also revealed a water molecule coordinated by the northern chain and ring of the toxin and residues of the lateral gate. This water coordination satisfies polar interactions of the northern chain and ring of the toxin that are exposed to the lipid environment. Thus, the number of contacts between oxygen atoms of the northern chain and ring of the toxin and oxygen atoms of water molecules was calculated for each isomer. Similar to interactions with the polar cluster residues, mycolactone B’s lipid exposed region was found to have more interactions with water molecules than mycolactone A (Figure 9F and Figure 10). Together, this difference in polar interactions likely contributes to a stronger binding affinity of the B isomer.

Next, we assessed the conformational changes of Sec61 induced by each isomer and found that the cytosolic side of the translocon adopts a more closed conformation with mycolactone B bound (Figure 11). This observation contradicts the conformational differences observed in our simulations using Gérard et al.’s [25] cryo-EM model, where the A-bound complex exhibited a relatively more closed lateral gate conformation. A more closed conformation of the translocon in Itskanov et al.’s [26] model was suggested to have implications regarding the difference in toxicity. Itskanov et al.’s [26] inhibitor-bound structures of Sec61 demonstrate the conformational plasticity of the binding pocket as the width of the lateral gate opening varies depending on the bound inhibitor. Among the seven inhibitor-bound structures, the lateral gate of the cotransin-bound structure was found to have the widest opening on the cytosolic side. It has been shown that cotransin is less effective in inhibiting the translocation of peptides that have a stronger targeting signal [31,32,33]. Itskanov et al. [26] proposed that the reason behind this observation could be the wider conformation of the lateral gate observed in their cotransin-bound structure, which might enable specific interactions between the lateral gate and a signal sequence, ultimately opening the lateral gate further and potentially causing the release of the inhibitor. Extending this reasoning to our results would suggest that a more closed conformation of the translocon, observed in our simulations, is associated with mycolactone B’s increased toxicity through a similar mechanism.

## 3. Discussion

Using all-atom MD simulations, we have herein probed the isomeric specificity of the macrolide exotoxin mycolactone, the sole causative agent in Buruli ulcer disease. Mycolactone is produced by *Mycobacterium ulcerans* in two isomeric forms (A and B in a 60:40 ratio), but only isomer B is thought to be cytotoxic [10]. Although the two isomers are separable by reversed-phase HPLC, they undergo rapid equilibration under standard laboratory conditions [34,35] and their separation within lipid bilayers has not been tested. Focusing on the primary source of the toxin’s cytotoxic effects, we looked for differences in how the A and B isomers associate with both the Sec61 translocon as well as the ER membrane. The latter was deemed relevant since it likely serves as a reservoir for the toxin prior to association with the translocon. This role is supported by early imaging studies that show uptake into the ER membrane within minutes of the toxin being exposed to cells [11], as well as mycolactone’s strong association with lipophilic carriers, including cellular membranes and HDL [13,14,15,36].

First, focusing on the ER membrane, we calculated free energy profiles for toxin association and permeation. Both isomers show a strong association with the ER membrane with energy minima corresponding to configurations that maximize polar and nonpolar interactions at the interface of the lipid headgroups and tails. However, isomer B has a stronger affinity for the ER membrane than isomer A (~−17 vs. −13 kcal/mol, respectively), with clear differences in the associated conformational ensembles (Figure 3 and Figure 4). The origin of this can be traced to the structure. Mycolactone A has a more collapsed structure, enabling more intramolecular interactions that stabilize the toxin in any environment. In contrast, mycolactone B is necessarily more extended, enabling and requiring more interactions with the environment. Given its amphiphilic nature, this leads to a greater stabilization of mycolactone B compared to A when the toxin transitions from water to the membrane, where both polar and nonpolar interactions are optimized.

The isomeric differences are additionally supported by the average interaction energies, which are stronger between the toxin and membrane for isomer B (more extended), and stronger for intramolecular interactions for isomer A (more compact) (Table 1). This also explains the broader angle distribution in the free energy profile of isomer A (Figure 3). More intramolecular interactions result in less angular dependence or need for polar interactions with the lipid head and/or glycerol groups. In contrast, mycolactone B’s extended conformation requires interactions with the lipid headgroups/glycerols to satisfy the polar tails, leading to a narrower angle distribution.

Collectively, isomer B has a stronger water-to-ER membrane association free energy due to its extended conformation; this enables more amphiphilic interactions with lipids. In comparison, isomer A is relatively more stable in water due to its intramolecular interactions. In recent work [15], we also demonstrated that isomer B preferentially associates with a model ER membrane containing POPC and POPE lipids over the more rigid model plasma membrane containing POPC, POPE, DPPC and cholesterol. The preferential association was driven by more favorable enthalpic interactions with water in the ER membrane and unfavorable disruption of lipid packing in the plasma membrane. The toxin pulled in the ER-lipids (POPC and POPE) in the plasma membrane, pushing away the saturated DPPC tails and cholesterol. Thus, we hypothesize that isomer B will preferentially localize to more disordered lipophilic carriers within the host, and that this could contribute to its localization to the ER. Our findings here extend this hypothesis to suggest that isomer B will be more likely to localize to the ER, and this increased localization in the membrane containing the Sec61 translocon could contribute to its toxicity via increased local population. An intriguing complication to this hypothesis is that increased affinity for the ER membrane will compete with association with the translocon. In other words, isomer B will have to be ~3 kcal/mol more stable in the translocon over isomer A in order to retain the same binding affinity for equivalent ER populations.

Turning to the Sec61 translocon, our simulations reveal clear differences in how each isomer associates with the protein in both Gérard et al.’s [25] and Itskanov et al.’s [26] cryo-EM models. Using Gérard et al.’s [25] structure, we found that mycolactone B has more interactions with a hydrophobic patch of residues that are thought to play a key role in signal peptide binding and subsequent peptide translocation [30]. The extended southern chain of isomer B enables better access to the binding site in order to interact with the hydrophobic patch region. In our analysis of Itskanov et al.’s [26] structure, mycolactone B was found to have interactions with the plug domain and both sides of the lateral gate of the channel, while mycolactone A only interacts with one side of the lateral gate. Mycolactone A’s lack of interactions with the plug and TM 7 reduce its ability to lock these gating elements in place to inhibit translocation. We also find that the B isomer inserts further into the pore constriction of the channel when compared to isomer A to form hydrophobic interactions and potentially block the pore from client proteins. Furthermore, key polar interactions of the toxin were found to be better satisfied with mycolactone B bound relative to A, which likely results in a stronger binding affinity of isomer B, making it less susceptible to being displaced by a strong signal sequence. Lastly, the complex bound with the B isomer adopts a relatively more closed conformation on the cytosolic side of the translocon, potentially making it more difficult for an incoming signal sequence to interact with the lateral gate for subsequent translocation.

The observed differences in binding sites between the two mycolactone-Sec61 cryo-EM structures may be a consequence of the different protein constructs, toxin concentrations and membrane-mimicking scaffolds used in the two studies. This possibility is interesting to consider in the context of the toxin’s association pathway coming from the ER membrane in vivo, which is more likely mimicked by canine microsomes [25] than non-lipidic peptidiscs [26]. However, it has also been proposed that both studies may have identified relevant binding sites for mycolactone [37] that act in distinct mechanisms of action during co-translational or post-translational translocation. The Sec61 translocon can function through either a co-translational process involving ribosomes or a post-translational process involving Sec62/63 [38,39,40]. Previous work has demonstrated that mycolactone exhibits different degrees of inhibition in these two translocation modes [41]. In co-translational translocation, the toxin exhibits a broad effect, inhibiting the translocation of various substrates. In contrast, its effect is more limited in post-translational translocation, affecting only a subset of substrates. Taken together, it is intriguing to consider that mycolactone might possess two distinct binding sites. In the site that is putatively involved in co-translational translocation (Gérard et al. [25]), our findings suggest that mycolactone acts by obstructing the hydrophobic patch residues from signal peptides. In the other site, potentially operating in post-translational translocation (Itskanov et al. [26]), the toxin locks the lateral gate and plug domains to prevent translocation dynamics.

While this study focuses on the most pathogenic mycolactone variant, mycolactone A/B, it is important to acknowledge the existence of several other variants. These include mycolactone C, produced by *M. ulcerans* in Australia; mycolactone D, found in *M. ulcerans* strains from China; mycolactone E, produced by *M. liflandii* in Sub-Saharan Africa; and mycolactone F, produced by *M. pseudo-shottsii* and certain strains of *M. marinum* worldwide [42,43,44,45,46,47,48,49]. Given the diverse range of mycolactone variants, there is a compelling opportunity for further research to explore their unique properties and effects, enabling a comprehensive comparison with mycolactone A/B in various contexts. It would be intriguing to investigate if other pathogenic variants such as mycolactone C exhibit similar interaction trends as mycolactone B, and whether non-pathogenic variants to humans, like mycolactone F, follow the same interaction trends as mycolactone A. Further investigation would provide valuable insights into the specific characteristics of each mycolactone variant and their potential implications.

## 4. Conclusions

Based on the simulations in this work, there are clear differences in the association mechanism of each mycolactone isomer with both the ER membrane and the Sec61 translocon, suggesting that both localization and association could play a role in isomer B’s increased cytotoxicity. Specifically, we find that isomer B has a greater affinity for the disordered ER membrane, which could lead to an increased local concentration around Sec61, thereby increasing its effective affinity for the translocon. Additionally, our findings reveal that mycolactone B has different interactions with and an impact on both proposed Sec61-toxin structures. It more readily interacts with the hydrophobic patch residues in Gérard et al.’s [25] model, potentially blocking signal peptide association. It also interacts with all three gating elements in the Itskanov et al. [26] model, supporting their proposed mechanism of inhibition in which the toxin blocks interaction with the signal peptide. This is supported by the more closed translocon conformation when mycolactone B is bound when compared to A, and, presumably, the isomer’s ability to block the translocation of peptides with a stronger targeting signal. Although the correct Sec61-toxin complex structure is not yet known, these findings may be informative in continued efforts to understand the complex mechanism of co-translational and post-translational protein translocation by the Sec61 translocon, as well as the potentially unique modes of inhibition. These insights are relevant to targeting specific interactions in the Sec61 translocon for therapeutic applications such as anticancer and antiviral treatment, as cancer cells and viruses rely on and exploit protein translocation into the ER to cause disease [50,51,52,53,54]. Additionally, our findings suggest that diagnostics targeting the mycolactone toxin with disordered lipophilic architectures should have an increased affinity for isomer B. Future experimental work testing these findings by quantifying the relative binding affinities of the two isomers to both the Sec61 translocon and different membrane compositions, as well as the inhibition levels of both isomers, will be important additions to the field.

## 5. Materials and Methods

The ER membrane was built with 198 units of 1-palmitoyl-2-oleoyl-sn-glycero-3-phosphocholine (POPC) and 102 units of 1-palmitoyl-2-oleoyl-sn-glycero-3-phosphoethanolamine (POPE) using the CHARMM-GUI Membrane Builder for Mixed Bilayers [55]. This phospholipid ratio was chosen to represent the composition of the ER membrane in mammals [56,57,58,59,60]. The lipid parameters were obtained from an Amber-based force field [61] with corrections to stabilize the hydrophilic and hydrophobic forces. TIP3P water molecules [62] were added to solvate the membrane with a hydration ratio of 60 molecules per lipid component. This created system boxes with 9.6 nm × 9.6 nm × 9.3 nm (*X*, *Y*, *Z*) that were energy minimized for 10,000 steps, heated until 310.15 K and then equilibrated for 250 ns until reached convergence. After the equilibration, a mycolactone molecule (one isomer per box) was randomly added to the water solution with arbitrary orientation. The toxin parameters (for each isomer) were the same developed and utilized by López et. al. and Aydin et al. [12,14]. For each isomer, three replicas were created and simulated.

The Newtonian equations of motion were integrated with a time step of 2 fs. During the equilibration and production, the pressure was kept constant at 1 bar (semiisotropically, as recommended for bilayer simulation) by the Berendsen barostat [63] with a coupling time every 5 ps. At the same time, the temperature was controlled at 310.15 K by the velocity rescaling thermostat (7) with a separately coupling time of 1 ps for each component (membrane + toxin and water). Using a cutoff distance of 1.0 nm, the short-range interaction list was updated every 10 steps, along with the long-range electrostatic interactions that were calculated using the smooth particle mesh Ewald method [64]. LINCS, the linear constraint solver, was applied to all hydrogen bonds.

The first set of mycolactone-Sec61 complexes employing Gérard et al.’s [25] structure were initiated from the equilibrated isomer-B complex embedded in a POPC membrane from the reported simulations [25] (PDB ID: 6Z3T) and simulated with previously reported parameters [12]. An initial structure of the mycolactone A complex was obtained by running a 10 ns-long MD run using parameters of a single bond in place of the toxin’s 4–5 double bond (Figure 1) to allow the toxin to convert to the A isoform. The mycolactone A bound to Sec61 system was then energy minimized and equilibrated using the same protocols employed in Gérard et al.’s [25] work. The POPC lipids were represented with the Slipids force field [65,66], and the Sec61 protein parameters were described by the Amber 99SB-ILDN force field [67]. TIP3P water molecules were again used to solvate the complexes. The MD production protocols were the same as those used by Gérard et al. [25] to perform consistent comparisons and analyses.

The second set of mycolactone-Sec61 complexes was built using Itskanov et al.’s [26] cryo-EM structure of the human Sec61 channel inhibited by mycolactone B (PDB ID: 8DO0). To obtain a complex with the A isomer, AutoGrid v4.2.6 and AutoDock v4.2.6 [68,69,70] were used to predict the binding mode of the toxin. AutoGrid was used to set up the toxin and protein and AutoDock was used to perform the docking simulations. Mycolactone B’s position on the protein in Itskanov et al.’s [26] structure was used as a reference to center the grid for docking. The grid was centered on coordinates: 66.249, 55.65, 55.241 Å (*X*, *Y*, *Z*) and then expanded to encompass surrounding residues to explore multiple potential binding sites in the region. The grid spacing was set to 0.260 Å, and the grid maps were adjusted to 86 × 92 × 94. Docking was carried out using a Lamarckian Genetic Algorithm (LGA), starting with an initial population of 150 random individuals. The LGA runs included a maximum of 27,000 generations with a maximum number of 2,500,000 energy evaluations. The mutation and crossover rates were set to 0.02 and 0.08, respectively, and an optional elitism parameter of 1 was employed to determine the number of top individuals that would survive into the next generation. The probability of performing a local search on an individual was 0.06 and a maximum of 300 iterations per local search was allowed. The maximum number of consecutive successes or failures before doubling or halving the search step was set to 4. In total, 10 LGA runs were performed. After the conformational search, the docked conformations were sorted based on energy. Finally, the coordinates of the lowest energy conformation were clustered using a root-mean-squared deviation of 2.0 Å. The lowest energy structure of mycolactone A which exhibited a similar binding orientation as mycolactone B in Itskanov et al.’s [26] structure was chosen to be modeled in MD.

Each isomer-Sec61 complex was then embedded in our model ER membrane using CHARMM-GUI Membrane Builder (Bilayer Builder) [55]. The POPC and POPE lipids were represented with the Slipids 2020 force field [71,72] and the Sec61 protein parameters were described by the Amber 99SB-ILDN force field [67]. TIP3P water molecules were again used to solvate the complexes. The toxin parameters were obtained from previous work [12,14,15]. Minimization and equilibration were carried out using a seven-step protocol suggested by CHARMM-GUI for the built system. The systems were energy minimized for 1000 steps of steepest descent energy minimization followed by 2000 steps of conjugate gradient energy minimization, using the steepest descent algorithm as the first step in both cases. A six-step equilibration was then carried out, which consisted of gradually removing position restraints on the lipids, protein, and toxin followed by a final equilibration run without any restraints. In all equilibration steps, the system was simulated for 100 ps using the NPT ensemble with temperature and pressure coupling at 310.15 K and 1 bar, respectively. Production was performed under the same conditions as described above.

All MD simulations (toxin with ER membrane or complexed with Sec61) were performed with the GROMACS 2019.4 package [73]. In-group developed Python scripts were used to calculate the hydration and obtain the two-dimensional Potential of Mean Force maps (2D-PMFs) for the membrane permeation process. MDAnalysis [74,75] was also used in Python scripts to perform contact analysis for the mycolactone-Sec61 complexes. VMD [76] was used to visualize the trajectories and generate the Figs. Finally, the GROMACS was patched with PLUMED 2.5.3 [77] to enable the employment of the enhanced method TTMetaD.

Membrane permeation is generally a slow and challenging process to be captured by MD simulations on accessible timescales. Enhanced samplings methods, such as Metadynamics [78,79], were developed to address this and allow the observation of this type of process. TTMetaD is an efficient variant of Metadynamics that enables the convergence of the system with only an approximated idea of the barrier surroundings, not its height [80]. This information can be obtained by unbiased simulations of the toxin permeation, e.g., in our previous studies [14,15]. In a simulation with MetaD, a bias energy is added to the Hamiltonian of the system through a small number of selected CVs that represent the slowest degrees of motion of the system. This bias energy is usually a Gaussian function that is centered at the previous visited configuration and represents the amount of energy necessary to sample the transition state in the CV space. In TTMetaD, the bias energy incremental rate decreases exponentially with respect to the local bias, first filling the transition state region, and only then converge through a more aggressive tempering of the Gaussian height [80,81]. Based on previous studies of small organic molecules [81,82] and specifically for mycolactone [14,15], two CVs are enough to describe the permeation process: the *z*-component distance of the center of mass of the lactone ring to the COM of the membrane as CV1; and the angle between the vector connecting the northern and southern chains’ hydroxyl groups and the normal vector to the bilayer. More detail can be seen in references [14,15].

As an efficient method for membrane permeation analysis [14,81,82], 2D PMFs were obtained to observe the energy distribution along the permeation process and to allow the calculation of the minimal free energy path (MFEP). The 2D-PMF is computed by the reverse of the average bias energy from the independent replicas followed by a diagonal symmetrization with respect to the CVs (toxin orientation and the center of the membrane). The MFEP was calculated using the zero-temperature string method [83] and represents the most probable path in the ensemble of permeation processes. One-dimensional (1D) free energy profiles were obtained by taking the average MFEP from the simulations for each isomer.

## Figures and Tables

**Figure 1 toxins-15-00486-f001:**
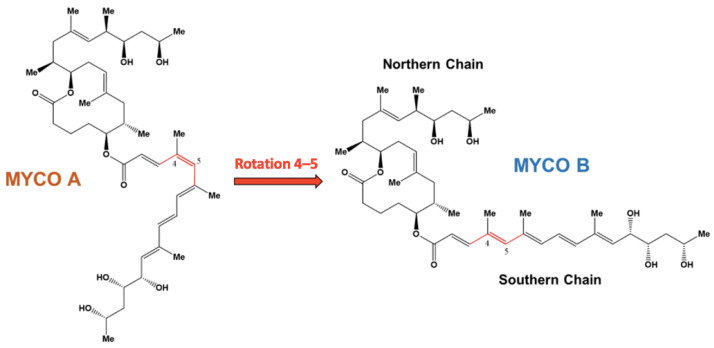
Structures of mycolactone A and B identifying the northern and southern chains. The dihedral angle 3–4–5–6 (cis/trans) is shown in red.

**Figure 2 toxins-15-00486-f002:**
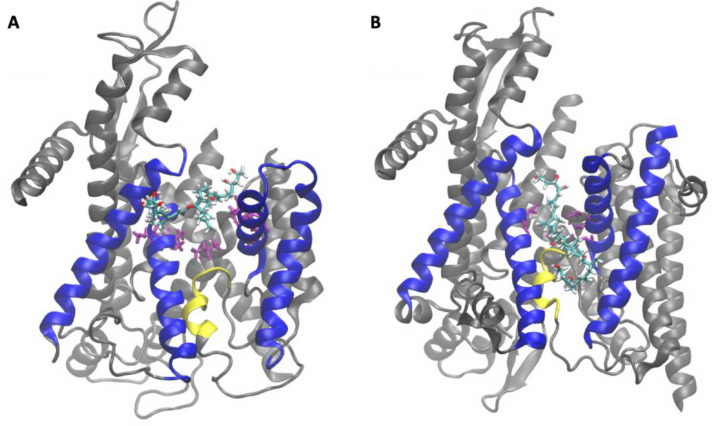
Structures of Sec61 inhibited by mycolactone reported by (**A**) Gérard et al. [25] (PDB ID: 6Z3T) and (**B**) Itskanov et al. [26] (PDB ID: 8DO0) highlighting mycolactone (colored by atom type), the plug (yellow), lateral gate (blue: TMs 2b, 3, 7 and 8) and pore ring residues (purple).

**Figure 3 toxins-15-00486-f003:**
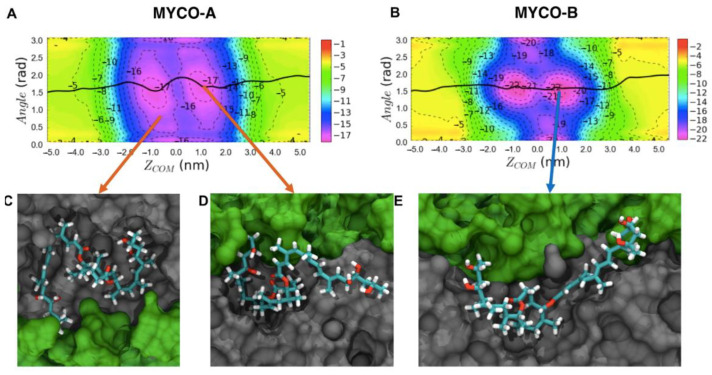
2D free energy profiles of the permeation of (**A**) mycolactone A and (**B**) mycolactone B through the ER membrane. The black lines trace the toxin’s most probable paths. (**C**–**E**) representative configurations. The toxin is colored by atom type, lipid headgroups, and tails are colored green and gray, respectively. Water is omitted for clarity. The membrane spans −2.0 < *Z_COM_* < 2.0 nm. The energy is shown in kilocalories per mole (kcal/mol).

**Figure 4 toxins-15-00486-f004:**
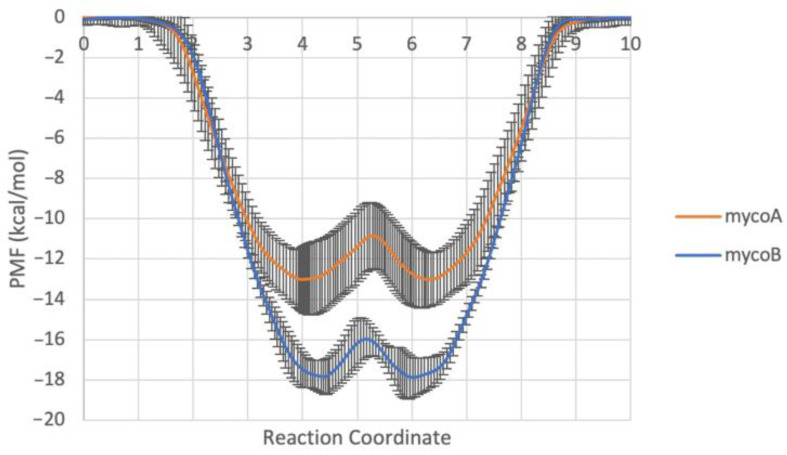
1D free energy profiles of each system along the minimum free energy path. The standard errors range from 0.03 to 1.82 kcal mol^−1^ for mycolactone A and from 0.06 to 1.22 kcal mol^−1^ for mycolactone B.

**Figure 5 toxins-15-00486-f005:**
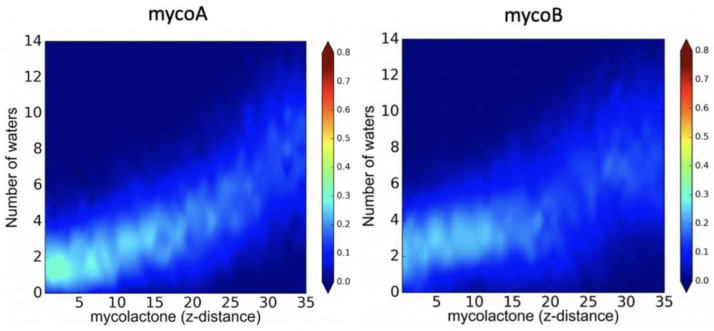
Probability distribution of the number of water molecules interacting with mycolactone with respect to the toxin position in the ER membrane in biased simulations. Z-distance is shown in angstroms.

**Figure 6 toxins-15-00486-f006:**
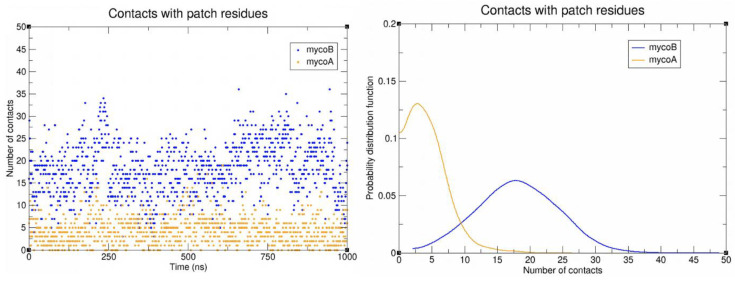
Number of contacts between mycolactone and patch residues as a function of time (**left**) and probability distribution function of the number of contacts between mycolactone and patch residues (**right**).

**Figure 7 toxins-15-00486-f007:**
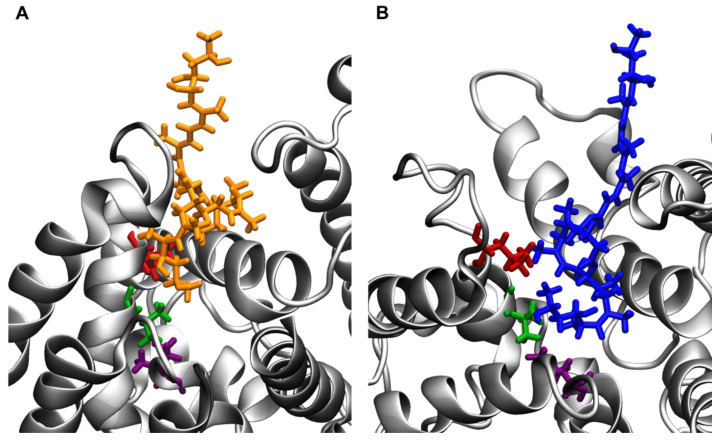
Position of (**A**) mycolactone A and (**B**) mycolactone B relative to patch residues. Isomer A and isomer B are shown in orange and blue, respectively. V85, L89, and I179 are shown in green, red, and purple, respectively, and Sec61 is shown in grey.

**Figure 8 toxins-15-00486-f008:**
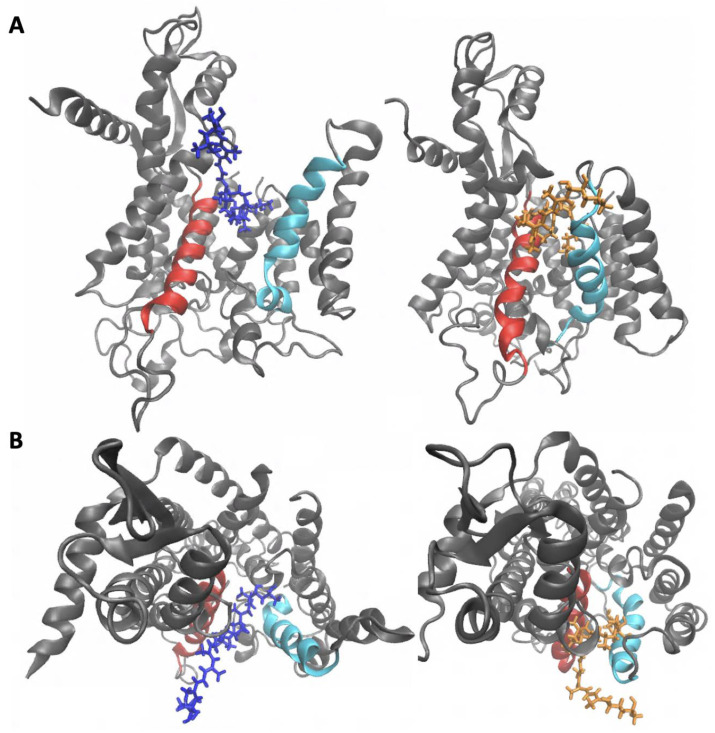
Configurations of mycolactone A/B-Sec61 complexes viewed from (**A**) the side, cutting through the membrane, and (**B**) above, looking down at the membrane surface. Mycolactone A (**left**) and B (**right**) are colored orange and blue, respectively, TMs 2 and 7 are colored cyan and red, respectively. The membrane is not shown for clarity.

**Figure 9 toxins-15-00486-f009:**
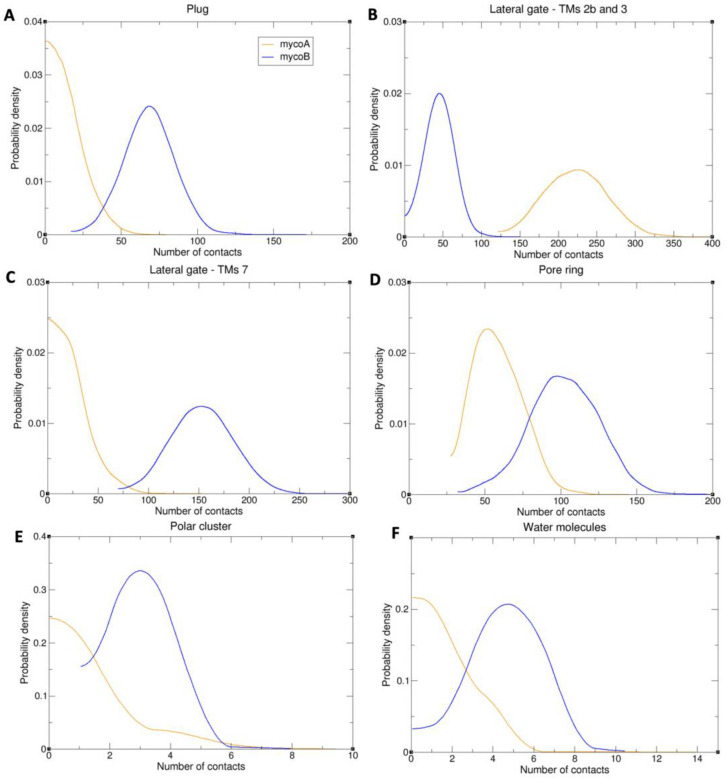
Probability of the number of contacts between mycolactone and the plug (**A**), TMs 2b, 3 (**B**), and 7 (**C**), and the pore ring (**D**), polar cluster residues (**E**), and water molecules (**F**).

**Figure 10 toxins-15-00486-f010:**
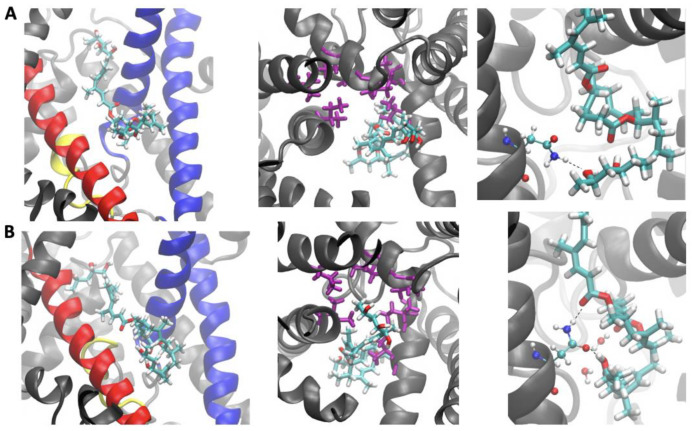
Positions of mycolactone A (**row A**) and mycolactone B (**row B**) relative to the plug and TMs 2b, 3 and 7 (**left**), the pore ring (**middle**) and N300 with coordinating water molecules (**right**). Mycolactone is shown as licorice sticks and colored by atom type. Sec61α is shown as cartoon ribbons with the plug, TMs 2b and 3, and TMs 7 colored yellow, blue, and red, respectively. The pore ring residues are shown as purple licorice sticks. N300 and water molecules are shown as balls and sticks and colored by atom type.

**Figure 11 toxins-15-00486-f011:**
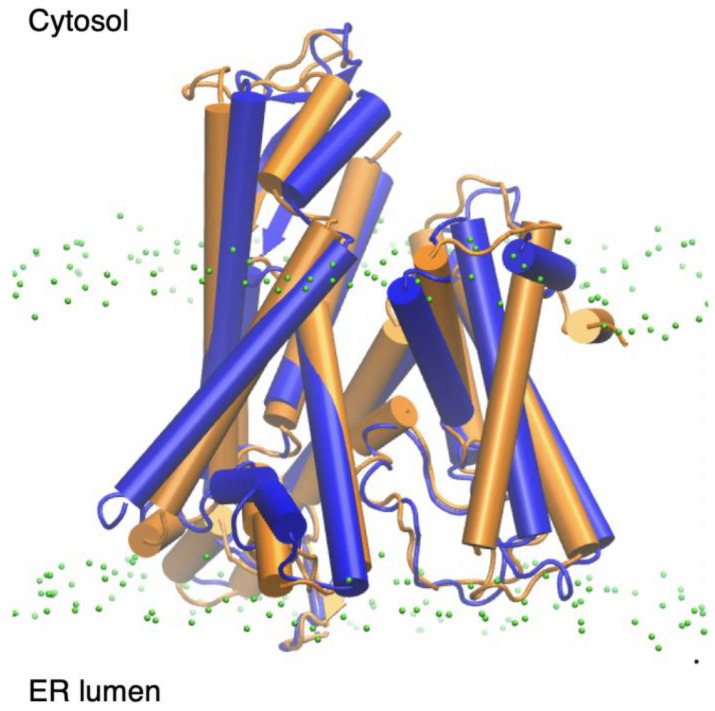
Conformations of Sec61 with mycolactone A (orange) and mycolactone B (blue) bound. Lipid head group phosphates are colored green.

**Table 1 toxins-15-00486-t001:** Average interaction potential energy of mycolactone with different components of the system.

Mycolactone Interaction Energies (kcal mol^−1^)				
	Lipids	Water	Intramolecular	Total
mycoA	−85.89 ± 0.32	−28.33 ± 0.36	−24.89 ± 0.16	−139.11
mycoB	−103.44 ± 0.43	−31.54 ± 0.36	−9.99 ± 0.19	−144.97

## Data Availability

Not applicable.

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
