# Peer review of "Mycolactone A vs. B: Multiscale Simulations Reveal the Roles of Localization and Association in Isomer-Specific Toxicity"

_toxins, 2023, doi:10.3390/toxins15080486_

Round 1

Reviewer 1 Report

The overall aim of this work was to compare and contrast the interaction of different mycolactone isomers with ER membranes and the Sec61 translocon. Mycolactone is a polyketide lactone, which self-equilibrates between two isoforms that isomerise around a double bond in one of the polyketide chains. The work is important, because the ER membrane likely concentrates mycolactone near translocons after it enters the cell to mediate its pathogenic effects during Mycobacterium ulcerans infection. The study consists of computational analysis of these interactions by means of simulations. The main conclusion is that mycolactone B is likely the most potent isoform, which supports several experimental conclusions from previous literature, although not all of these are cited.

General concept comments (major concerns):

The main concern relates to the focus on one of two structures of Sec61 inhibited by mycolactone, reference 22 by Itsnakov and colleagues (a peer reviewed version now published: PMID 37169959). This work, while important, used an unusual method to stabilise translocons in an open state by expressing a hybrid translocon (comprised of human transmembrane regions and yeast extracellular regions), combining it with other translocon-associated proteins Sec62 and Sec63, and then integrating this into a non-lipid peptidisc. Inhibitors were added at extremely high concentrations to these complexes post hoc. While this allowed for extremely high resolution cryo-EM structures, there is some debate about whether the inhibited Sec61 structures accurately represents the physiological ones. Contemporaneous or earlier Cryo-EM structures of Sec61 in ribosome-translocon complexes (RTCs), inhibited by mycolactone (PMID 32692975), CADA (PMID 36867692) and the co-transin analogue KEZAR-8445 (PMID 37169961) all have similar structures, despite each study being performed entirely independently, and showing the inhibitor bound at a different site. Notably the positioning of both CADA and mycolactone differ between the hybrid-peptidisc structures and the RTCs. In each case, the structure of Sec61 differs from ref 22.

Therefore, a major weakness of the work is that it does not consider the alternative (and potentially more biologically relevant) model of inhibited Sec61, published in 2020. Indeed, mycolactone A and B have been docked on to this structure in the past (PMID 34726690). The work in Section 2.2 (line 227 onwards) should repeat this analysis with PDB 6Z3T, to confirm that the conclusions are relevant in both models. Since previous work also performed docking or MD simulations and presented experimental evidence for preferential binding of mycolactone B to RTC, the discussion of the findings in this interesting paper would be greatly enhanced by incorporating all relevant references.

Specific comments (minor concerns):

·         Line 82: Ref 1 should also be cited here

·         Line 139: Justify the use of 198 POPC: 102 POPE as the correct composition and ratio of lipids for the ER.

·         Line 310: Formatting error. It would also be worth pointing out that is isn’t know if it’s possible for the two isoforms to be separated within lipid bilayers.

·         Line 415: the PDB of the structure(s) used should be given.

Author Response

Introductory Comment: The overall aim of this work was to compare and contrast the interaction of different mycolactone isomers with ER membranes and the Sec61 translocon. Mycolactone is a polyketide lactone, which self-equilibrates between two isoforms that isomerise around a double bond in one of the polyketide chains. The work is important, because the ER membrane likely concentrates mycolactone near translocons after it enters the cell to mediate its pathogenic effects during Mycobacterium ulcerans infection. The study consists of computational analysis of these interactions by means of simulations. The main conclusion is that mycolactone B is likely the most potent isoform, which supports several experimental conclusions from previous literature, although not all of these are cited.

General concept comments (major concerns): The main concern relates to the focus on one of two structures of Sec61 inhibited by mycolactone, reference 22 by Itsnakov and colleagues (a peer reviewed version now published: PMID 37169959). This work, while important, used an unusual method to stabilise translocons in an open state by expressing a hybrid translocon (comprised of human transmembrane regions and yeast extracellular regions), combining it with other translocon-associated proteins Sec62 and Sec63, and then integrating this into a non-lipid peptidisc. Inhibitors were added at extremely high concentrations to these complexes post hoc. While this allowed for extremely high resolution cryo-EM structures, there is some debate about whether the inhibited Sec61 structures accurately represents the physiological ones. Contemporaneous or earlier Cryo-EM structures of Sec61 in ribosome-translocon complexes (RTCs), inhibited by mycolactone (PMID 32692975), CADA (PMID 36867692) and the co-transin analogue KEZAR-8445 (PMID 37169961) all have similar structures, despite each study being performed entirely independently, and showing the inhibitor bound at a different site. Notably the positioning of both CADA and mycolactone differ between the hybrid-peptidisc structures and the RTCs. In each case, the structure of Sec61 differs from ref 22.

Therefore, a major weakness of the work is that it does not consider the alternative (and potentially more biologically relevant) model of inhibited Sec61, published in 2020. Indeed, mycolactone A and B have been docked on to this structure in the past (PMID 34726690). The work in Section 2.2 (line 227 onwards) should repeat this analysis with PDB 6Z3T, to confirm that the conclusions are relevant in both models. Since previous work also performed docking or MD simulations and presented experimental evidence for preferential binding of mycolactone B to RTC, the discussion of the findings in this interesting paper would be greatly enhanced by incorporating all relevant references.

Author reply 1: We would like to thank the reviewer for bringing this to our attention.  We actually started this project focusing on PDB 6Z3T. We were then advised to focus on the new structure, but were unaware of the noted limitations. We are happy to include a more thorough analysis, comparing results for both structures. Modifications have been made throughout the manuscript (tracked in blue font).

Specific comments (minor concerns):

Line 82: Ref 1 should also be cited here

Author reply: We have included the citation as recommended.

Line 139: Justify the use of 198 POPC: 102 POPE as the correct composition and ratio of lipids for the ER.

Author reply: We have included in Materials and Methods references (experimental and computational) indicating the ER membrane should be represented with a ratio of approximately 2:1 POPC:POPE.

"This phospholipid ratio was chosen to represent the composition of the ER membrane in mammals [45-49]"

Line 310: Formatting error. It would also be worth pointing out that is isn’t know if it’s possible for the two isoforms to be separated within lipid bilayers.

Author reply: We fixed the formatting and added more context about the impossibility of isolating the two isomers.

"Although mycolactone is produced by Mycobacterium ulcerans in two isomeric forms (A and B in a 60:40 ratio), only isomer B is thought to be cytotoxic [10]. Although the two isomers are separable by reversed-phase HPLC they undergo rapid equilibration under standard laboratory conditions [31, 32] and their separation within lipid bilayers has not been tested."

Line 415: the PDB of the structure(s) used should be given.

Author reply: We have added a new paragraph with the information regarding the new structure, including PDB IDs for both structures.

Methods:

           "The first set of mycolactone-Sec61 complexes employing Gérard et al.’s [21] structure was built based on the models described and used by the authors [21] (PDB ID: 6Z3T). Each isomer-complex replica was created using an equilibrated structure of mycolactone B bound to Sec61 embedded in a POPC membrane, replacing the isomer-B structure and topology parameters [12] when needed to simulate mycolactone A. An initial structure of the mycolactone A complex was obtained by running a 10 ns-long MD run using parameters of a single bond in place of the toxin’s 4-5 double bond (Figure 1) to allow the toxin to convert to the A isoform. The mycolactone A bound to Sec61 system was then energy minimized and equilibrated using the same protocols employed in Gérard et al.’s [21] work. The POPC lipids were represented with the Slipids force field [54, 55], and the Sec61 protein parameters were described by the Amber 99SB-ILDN force field [56]. TIP3P water molecules were again used to solvate the complexes. The MD production protocols were the same as those used by Gérard et al. [21] to perform consistent comparisons and analyses.

            The second set of mycolactone-Sec61 complexes was built using Itskanov et al.’s [22] cryo-EM structure of the human Sec61 channel inhibited by mycolactone B (PDB ID: 8DO0)."

Reviewer 2 Report

This article discuss about the possible reasons of two different isomers a natural product from molecular level.

The methods the authors chose the traditional molecular dynamic computation method as well as some related verification methods.

The results sound reasonable and interesting in explanation the inner subtile different toxicity properties between two isomers.

It is suggested to be published before the following several questions are pondered again for clarity reasons. 

1. The title should be changed into a positive sentence instead of the question style. i.e. (only as a reference)

The possible reasons for the higher toxicity of Mycolactone B than its A isomer from the expect of delocalization and association via molecular modeling.

2. The distance between oxygens was used in evaluation the interaction between the toxin and water molecule to calculate the probability distribution of number of water molecules in contact with toxin on page 6.

Since the contact was suggested as a kind of polar interaction, especially hydrogen bond, the distance should be changed into that distance between oxygen and hydrogen. 

Some minor grammatical problem should be payed attentions.

for examples,

1. The expression styles of "of ...... of ........" were used too often. 

2. Some sentences should be made as succinct as possible.

like on page 6, Line 228-229-230, the one set of sentence should use one object and verb except for those subordinations. 

Author Response

Introductory comment: This article discuss about the possible reasons of two different isomers a natural product from molecular level.

The methods the authors chose the traditional molecular dynamic computation method as well as some related verification methods.

The results sound reasonable and interesting in explanation the inner subtile different toxicity properties between two isomers.

It is suggested to be published before the following several questions are pondered again for clarity reasons.

Comment 1: The title should be changed into a positive sentence instead of the question style. i.e. (only as a reference)

The possible reasons for the higher toxicity of Mycolactone B than its A isomer from the expect of delocalization and association via molecular modeling.

Author reply 1: This is a good suggestion. We have changed the title to:

"Mycolactone A vs. B: Exploring the role of localization and association in isomer-specific toxicity"

Comment 2: The distance between oxygens was used in evaluation the interaction between the toxin and water molecule to calculate the probability distribution of number of water molecules in contact with toxin on page 6.

Since the contact was suggested as a kind of polar interaction, especially hydrogen bond, the distance should be changed into that distance between oxygen and hydrogen.

Author reply 2: Thank you for pointing this out. Although this is technically feasible, in practice hydrogen bonds are typically quantified by donor-acceptor distances of ≤3 angstroms. Thus, for the purposes of our study, we believe this definition is most appropriate.

Comments on the Quality of English Language: Some minor grammatical problem should be payed attentions.

for examples,

  1. The expression styles of "of ...... of ........" were used too often.

  1. Some sentences should be made as succinct as possible.

like on page 6, Line 228-229-230, the one set of sentence should use one object and verb except for those subordinations.

Author reply: We have rephrased this set of sentences to be more concise (Lines 285-288).

"Given that the toxin’s cytotoxic effects have been strongly linked to its interaction with the Sec61 translocon, we next evaluated differences in the association of the two isomers.  Both Gérard’s [21] and Itskanov’s [22] cryo-EM structures were evaluated since it is not yet known which is biologically more relevant."

Reviewer 3 Report

The authors focused on Mycolactone A/B and found that mycolactone B is probably the one able to interact with Sec61 channel and ER membrane.

I recommend to at least discuss the probable role of Mycolactone A in other cellular effect of the toxin.

I also recommend to perform same simulations with other mycolactones (C, D, E...) known to be produced by M. ulcerans species, which are also able to induce M. ulcerans infection.

Author Response

General Comment: The authors focused on Mycolactone A/B and found that mycolactone B is probably the one able to interact with Sec61 channel and ER membrane.

I recommend to at least discuss the probable role of Mycolactone A in other cellular effect of the toxin.

Author reply: We have included a more in-depth discussion on the other cellular effects of the toxin (Lines 84-93).

"Mycolactone acts by invading host cells and interacting with multiple intracellular targets, resulting in various cellular effects. One such target is the Angiotensin II receptor (AT2R), a key player in signal transduction among neurons responsible for pain perception. By binding to this receptor, the toxin initiates a signaling cascade that ultimately leads to hyperpolarization of neurons, impairing their ability to transmit pain signals and resulting in analgesia [17, 18].Another mycolactone target is the Wiskott-Aldrich syndrome protein (WASP). WASP is a regulatory protein that plays a crucial role in actin filament polymerization, a process vital for the cytoskeleton's structural integrity and functions such as movement and adhesion. Upon binding to WASP, mycolactone triggers its constant activation, causing uncontrolled growth of actin filaments, impairing cell adhesion, and leading to the development of skin ulcers [2, 19]."

I also recommend to perform same simulations with other mycolactones (C, D, E...) known to be produced by M. ulcerans species, which are also able to induce M. ulcerans infection.

Author reply: This is an interesting recommendation. To the best of our knowledge, mycolactone C is the only other variant that is pathogenic. However, it is significantly less pathogenic than mycolactone B. Although it would be interesting to see if mycolactone C follows the same association trends as mycolactone B, it is beyond the scope of the current study as all simulations and simulation time would essentially have to be repeated. This would take many months and phenomenal computing time. Since isomer B is the primary virulence factor r, we believe that comparing isomers A and B is sufficient for the purposes of this study.